

# The value of dual-energy computed tomography (DECT) in the diagnosis of urinary calculi: a systematic review and meta-analysis of retrospective studies

Peipei Feng, Guochao Li and Peng Liang

Department of Imaging, Yantaishan Hospital, Yantai, China

## ABSTRACT

**Objective**. Dual-energy computed tomography (DECT) imaging technology opens a new idea and method for analyzing stone composition, which can obtain several quantitative parameters reflecting tissue-related information and energy images different from traditional images. However, the application of DECT in diagnosing urinary calculi remains unknown. This study aims to evaluate the value of DECT in diagnosing urinary calculi by meta-analysis.

**Methods**. PubMed, EMBASE, Web of Science, and the Cochrane Library were searched to articles published from the establishment of the databases to April 18, 2023. We reviewed the articles on the diagnosis of urinary calculi detected by DECT, established standards, screened the articles, and extracted data. Two researchers carried out data extraction and the Cohen's unweighted kappa was estimated for inter-investigator reliability. The quality of the literature was evaluated by the diagnostic test accuracy quality evaluation tool (QUADAS-2). The heterogeneity and threshold effects were analyzed by Meta-Disc 1.4 software, and the combined sensitivity, specificity, positive likelihood ratio, negative likelihood ratio, and diagnostic ratio were calculated. The combined receiver-operating characteristic (ROC) curve was drawn, and the value of DECT in the diagnosis of urinary calculi was evaluated by the area under the curve (AUC). The meta-analysis was registered at PROSPERO (CRD42023418204).

**Results**. One thousand and twenty-seven stones were detected in 1,223 samples from 10 diagnostic tests. The analyzed kappa alternated between 0.78-0.85 for the document's retrieval and detection procedure. The sensitivity of DECT in the diagnosis of urinary calculi was 0.94 (95% CI [0.92–0.96]). The positive likelihood ratio (PLR) of DECT in the diagnosis of urinary stones was 0.91 (95% CI [0.88–0.94]), and the negative likelihood ratio (NLR) was 0.08 (95% CI [0.05–0.11]). The specificity of DECT for detecting urinary calculi was 0.91 (95% CI [0.88–0.94]). The area under the curve of the summary receiver operator characteristic (SROC) was 0.9875. The sensitivity of dual-energy CT in the diagnosis of urinary calculi diameter <3 mm was 0.94 (95% CI [0.91–0.96]). The PLR of DECT in the diagnosis of urinary stones diameter <3 mm was 10.79 (95% CI [5.25 to 22.17]), and the NLR was 0.08 (95% CI [0.05–0.13]). The specificity of DECT for detecting urinary calculi <3 mm was 0.91 (95% CI [0.87–0.94]). The SROC was 0.9772.

**Conclusion**. The DECT has noble application value in detecting urinary calculi.

**Subjects** Radiology and Medical Imaging, Urology

Corresponding author
Peng Liang, liang20220220@126.com

**Keywords** Dual-energy computed tomography, QUADAS-2, Urinary calculi, Area under the curve, Heterogeneity

# INTRODUCTION

Urolithiasis is a common cause of colicky pain and urinary tract obstruction (*Quhal & Seitz, 2021*). The clinical treatment of urolithiasis differs according to the stone components and thus the identification of these components before treatment would assist both the clinical treatment and prognosis of patients (*Bartges & Callens, 2015*). Conventional nonenhanced CT (CNCT) is currently the gold standard for diagnosing urinary calculi. However, some patients still need CT urography (CTU) to evaluate urinary obstruction and bilateral renal function after CNCT, resulting in increased radiation dose and potential nephrotoxicity, and limiting the application of CTU (*Brisbane, Bailey & Sorensen, 2016*). Dual-energy computed tomography (DECT) imaging provides a novel direction and method for the analysis of stone composition, resulting in the acquisition of several quantitative parameters that reflect tissue-related information and energy images that differ from those obtained by traditional imaging and which can be used for stone composition analysis (*Murray et al., 2019*). DECT uses high (140 kV) and low (80 kV, 100 kV) energy for synchronous scanning, allowing the acquisition of original CT values using two kinds of energy. The two sets of data allow simulation of the attenuation coefficient of matter in the energy range of 40 −190 keV, and the acquisition of single-energy images of 151 groups of keV and the curve of the energy spectrum (*Kaza et al., 2017*).

Previous studies have found that enhanced scanning using dual-energy CT in the diagnosis of urinary calculi could reduce the scanning times and the effective radiation dose by 20% to 50% (*Megibow, Kambadakone & Ananthakrishnan, 2018*). Both retrospective and prospective cohort studies on the use of DECT for the detection of urinary calculi have shown high diagnostic sensitivity and specificity (*Kordbacheh et al., 2019*; *Lazar et al., 2020*), although others have reported disappointing results (*Mansouri et al., 2015*; *Ahn, Oh & Seo, 2015*). Thus, there is no consensus on the accuracy of DECT in diagnosing urinary calculi and whether it can be used effectively in clinics. Thus, an elucidation of whether the diagnostic accuracy of DECT is sufficient for detecting urinary calculi is necessary. In this meta-analysis, we systematically evaluated the accuracy of DECT in the detection of urinary calculi from the point of view of evidence-based medicine through meta-analysis, using CNCT as the gold standard.

# MATERIALS & METHODS

## Inclusion and exclusion criteria

The inclusion criteria for the meta-analysis were studies that: evaluated patients with suspected urinary calculi, with no restrictions on age, sex, ethnicity, and country; used DECT in the diagnosis of urinary calculi; used CNCT as the diagnostic gold standard; allowed the extraction of data on a 2 × 2 table, such as true positive, false positive, false negative, and true negative.

The exclusion criteria were: use of DECT for non-diagnostic purposes; *in vitro* studies, animal experiments, reviews, case reports, letters, conference papers, or comments; articles not written in English; presence of >4 "No" or "Unclear" results in the QUADAS 2 evaluation; lack of a full-text version.

## Search approach

In the population, intervention, comparison, and outcomes (PICO) design used for the preparation of the forefront inquiry of the systematic review, the research question was as follows: "To determine whether DECT is effective in the detection of urinary stones in adult patients with urinary calculi, in comparison with routine tests". The PubMed, EMBASE, Web of Science, and Cochrane Library databases were searched from their establishment to April 18, 2023, to identify relevant articles. The search expressions "dual-energy computed tomography", "dual-energy CT", "DECT", "urinary calculi", "urinary calculus", "urinary tract stones", "urinary stone", "renal stone", "kidney stone", "urolithiasis", and "nephrolithiasis" were used. The search strategy included both free-text words and medical subject heading (MeSH) terms. Two researchers searched the databases independently. The study was approved by the Institutional Review Board and Research Ethics Committee of the Yantaishan Hospital. The meta-analysis was registered at PROSPERO (CRD42023418204).

## Quality evaluation

The quality evaluation of the articles was completed independently by two researchers (PPF and PL). A third reviewer made the final decision, confirming that at least two examiners had assessed each identified article. Discussions were held frequently throughout the evaluation procedure to reach consensus on specific cases; this method was used to unify assessment rules. The Quality Assessment of Diagnostic Accuracy Studies (QUADAS-2) (*Zieliński, Zieba & Byś, 2021*) was used to evaluate the research quality of the studies included in the meta-analysis. After discussion and consensus, Review Manager 5.2 software was used to evaluate QUADAS-2. The primary (most important) outcome of the review was the diagnostic performance of dual-energy CT in the detection of urinary calculi, measured using the area under the curve (AUC) of the summary receiver operating characteristic (SROC) curve.

## Statistical analysis

Stata (version 12.0) and Meta-DiSc software (version 1.4) were used to assess the heterogeneity and threshold effect of the included articles, and sensitivity analysis was performed. Heterogeneity analysis was conducted using the Cochran-Q and $I^2$ tests, where $I^2$ >50% suggested apparent heterogeneity and a bivariate random effect model was used for the combined analysis. Values of $I^2$ <50% suggested that the fixed effect model should be used for the combined analysis. The presence of a threshold effect was evaluated through the shape of the ROC curve and the Spearman correlation coefficient; when there was a threshold effect, only the area under the combined ROC curve was calculated, while if there was no threshold effect, the source of heterogeneity of the non-threshold effect was analyzed. Subgroup and sensitivity analyses were performed, and the combined sensitivity,
specificity, positive likelihood ratio (PLR), negative likelihood ratio (NLR), diagnostic ratio, and 95% confidence interval (CI) were calculated. Deek's funnel chart was drawn by Stata 15 software to evaluate whether there was publication bias. In the subgroup analysis, $P$-values <0.05 suggested that the source of the heterogeneity between studies was related to a covariable, and the relative diagnostic odds ratio (RDOR) was calculated after excluding covariates from high to low.

## RESULTS

### Characteristics and selection of the included studies

Figure 1 illustrates the search process. The initial searches of the online databases identified 225 potentially relevant studies. Of these, 136 were found to be duplicates and, together with 24 irrelevant articles, were excluded. After reading the full texts of the studies, a further 44 were found not to satisfy the inclusion criteria, and these together with 11 studies with insufficient data for construction of the contingency tables, were excluded. Ultimately, 10 articles were included in the meta-analysis. The characteristics of the included articles are shown in Table 1.

### Risk-of-bias and quality assessment

The data from the studies were extracted, analyzed, and evaluated by QUADAS-2. Figure 2 showed the assessment of bias risk. A high risk of index text bias was detected in six studies. Thus, the quality of the articles met the requirements of the meta-analysis.

### Quantitative synthesis

Significant heterogeneity in the sensitivity, specificity, and PLR of DECT in the diagnosis of urinary stones ($I^2 = 74.5\%$, $62.2\%$, $70.1\%$, all $P < 0.1$) was found in the included studies. Using the random effects model, the sensitivity of DECT in the diagnosis of urinary calculi was found to be 0.94 (95% CI [0.92–0.96], Fig. 3), with a PLR of 0.91 (95% CI [0.88–0.94], Fig. 4) and an NLR of 0.08 (95% CI [0.05–0.11], Fig. 5). The specificity of DECT in detecting urinary calculi was 0.91 (95% CI [0.88–0.94], Fig. 6). The AUC under the SROC curve was 0.9875, demonstrating excellent diagnostic performance (Fig. 7). Furthermore, there was no significant heterogeneity in the sensitivity, specificity, PLR, and NLR of DECT in the diagnosis of urinary stones with diameters <3 mm($I^2 = 16.3\%$, $0\%$, $0\%$, $29.5\%$). Using the fixed effects model, the sensitivity of DECT in the diagnosis of urinary calculi with diameters <3 mm was 0.94 (95% CI [0.91–0.96], Fig. 8) with a PLR of 10.79 (95% CI [5.25 to 22.17], Fig. 9) and an NLR of 0.08 (95% CI [0.05–0.13], Fig. 10). The specificity of DECT for detecting urinary calculi with diameters <3 mm was 0.91 (95% CI [0.87–0.94], Fig. 11). The AUC under the SROC curve was 0.9772, indicative of superior diagnostic performance (Fig. 12).

### Subgroup analysis

Forest plots showed that the $I^2$ value for the overall sensitivity was 74.5%, which described the heterogeneity. Subgroup analysis was conducted on the basis of the CT equipment, voltage, and article quality. The results of the analysis showed that heterogeneity between

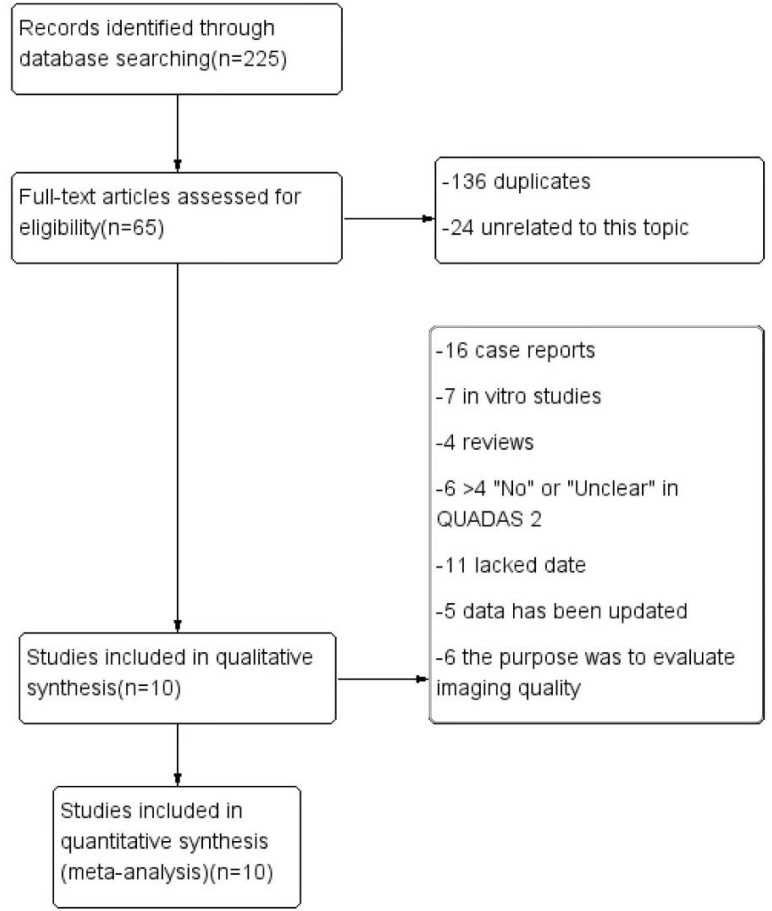

**Figure 1    The flow chart of literature extraction.**

studies was unrelated to CT equipment, tube voltage, and study quality (all *P* > 0.05, Table 2).

## DISCUSSION

Urolithiasis is a common disease that is frequently observed in clinics, and its incidence is increasing. There are significant differences in the clinical treatment of stones with different components, and thus evaluating the stone composition before treatment would assist both the clinical treatment and prognosis of patients. The recurrence rate of urolithiasis is high, and patients often need to undergo partial exclusion diagnostic examination (except for urinary neoplastic lesions), resulting in repeated examination (*Schönberg, Budjan & Hausmann, 2018*). In recent years, CT urography (CTU) has been widely used in examinations of urinary calculi. The overall scanning time of CTU is long, the radiation dose is high, and there is potential nephrotoxicity. Therefore, there is a need for radiologists to reduce the radiation dose while obtaining high-quality images and meeting the needs of the patients (*Cruz et al., 2019*). The overall scanning time of CTU is long, the radiation

**Table 1  Characteristics of all studies involved in this meta-analysis.**

| Study | Country | Number of patients | CT (kV) | Number of stones | Study center |
|---|---|---|---|---|---|
| Jepperson et al. (2013) (*Lazar et al., 2020*) | USA | 89 | 80,140 | 27 | Department of Radiology, Mayo Clinic, USA |
| Elisabeth et al. (2021) (*Mansouri et al., 2015*) | Germany | 70 | 80,140 | 34 | Department of Diagnostic and Interventional Radiology, Medical Faculty, University Dusseldorf, Germany |
| Mehmet et al. (2023) (*Ahn, Oh & Seo, 2015*) | Turkey | 115 | 100,140 | 83 | Department of Radiology, Faculty of Medicine, İnönü University, Malatya, Turkey |
| Botsikas et al. (2014) (*Zieliński, Zieba & Byś, 2021*) | Switzerland | 116 | 80,140 | 23 | Radiology Department, Geneva University Hospital, Geneva, Switzerland |
| Katherine et al. (2022) (*Botsikas et al., 2014*) | Australia | 1760 | 80,140 | 1740 | Faculty of Medicine and Health, University of Sydney, Camperdown, Australia |
| Jung et al. (2016) (*Schönberg, Budjan & Hausmann, 2018*) | Korea | 296 | 80/100,140 | 148 | Department of Radiology, Samsung Medical Center, Sungkyunkwan University School of Medicine, Seoul |
| Ascenti et al. (2016) (*Cruz et al., 2019*) | Italy | 84 | 80,140 | 75 | Department of Radiological Sciences, University of Messina, Messina, Italy |
| Chen et al. (2016) (*Hamid et al., 2021*) | China | 171 | 80,140 | 45 | Department of Medical Imaging, Kaohsiung Medical University Hospital, Kaohsiung, Taiwan |
| Karlo et al. (2013) (*Tatsugami et al., 2022*) | Switzerland | 100 | 80,140 | 104 | Department of Radiology, University Hospital Zurich, Switzerland |
| Moon et al. (2014) (*Sajja et al., 2020*) | Korea | 29 | 100,140 | 62 | The Department of Radiology and Centre for Imaging Science, Sungkyunkwan University School of Medicine, Seoul |

dose is high, and there is nephrotoxicity. Therefore, it is the direction of radiologists to reduce the radiation dose while obtaining high-quality images and meeting the needs of clinical and patients.

DECT imaging uses two sets of mutually perpendicular ball tubes and detectors in the rotating frame, with two tube voltages of 140 kV and 80 kV 140 kV. After data conversion, the discs are transformed by enhanced scanning (*Hamid et al., 2021*). The virtual single-energy spectrum imaging by DECT can draw different contrast-to-noise ratio (CNR) curves by

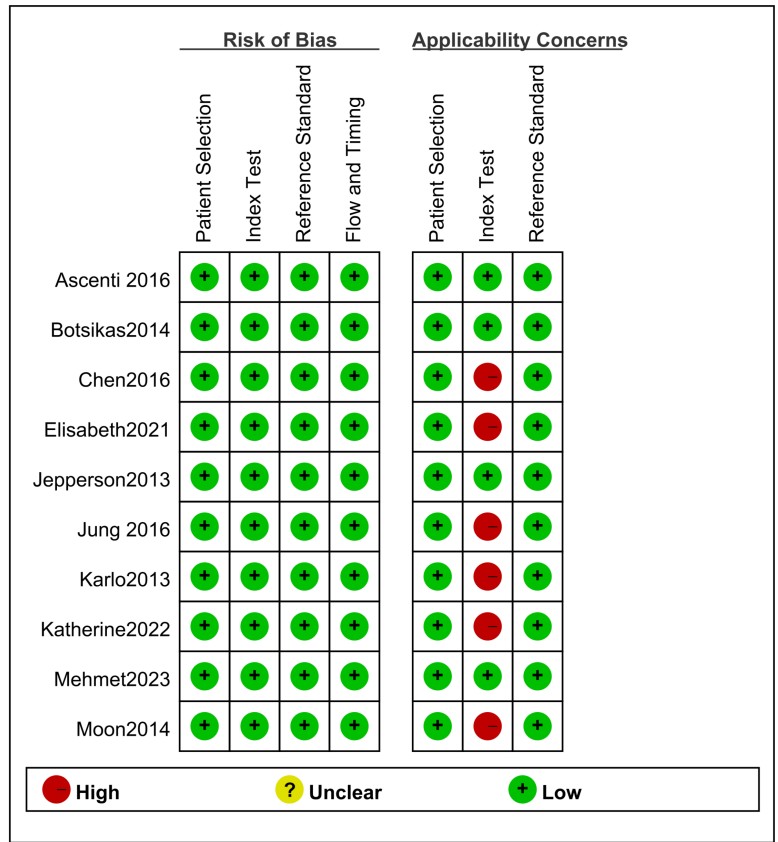

**Figure 2** The results of quality evaluation on included literature.

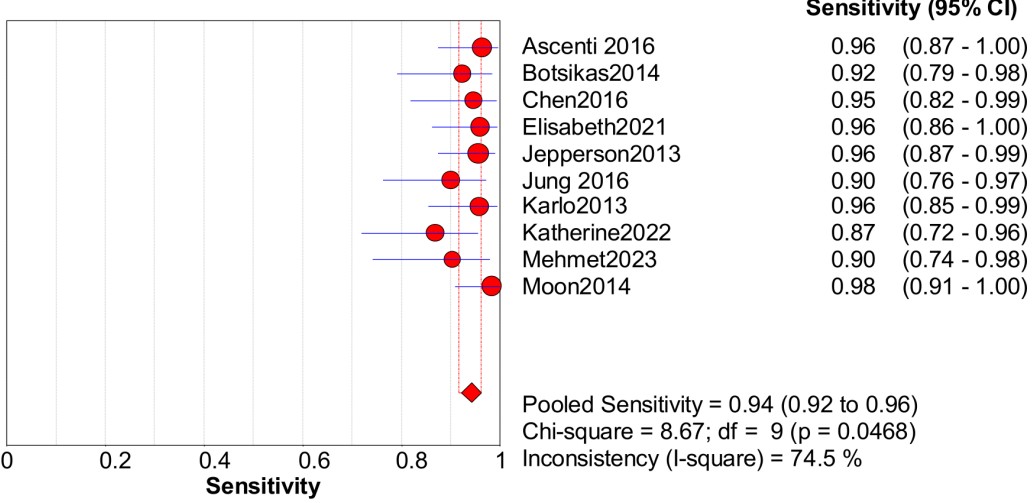

**Figure 3** Forest map of the sensitivity of dual energy CT in the detection of urinary calculi.

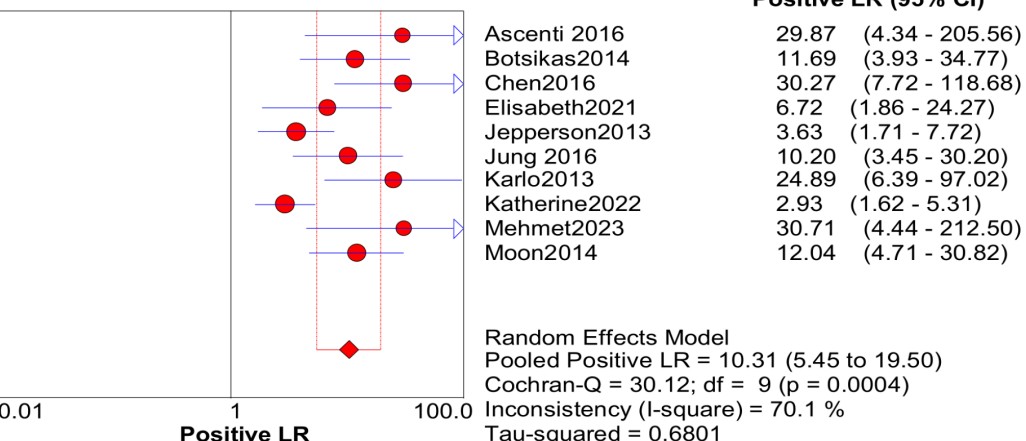

**Figure 4** Forest map of the PLR of dual energy CT in the detection of urinary calculi.

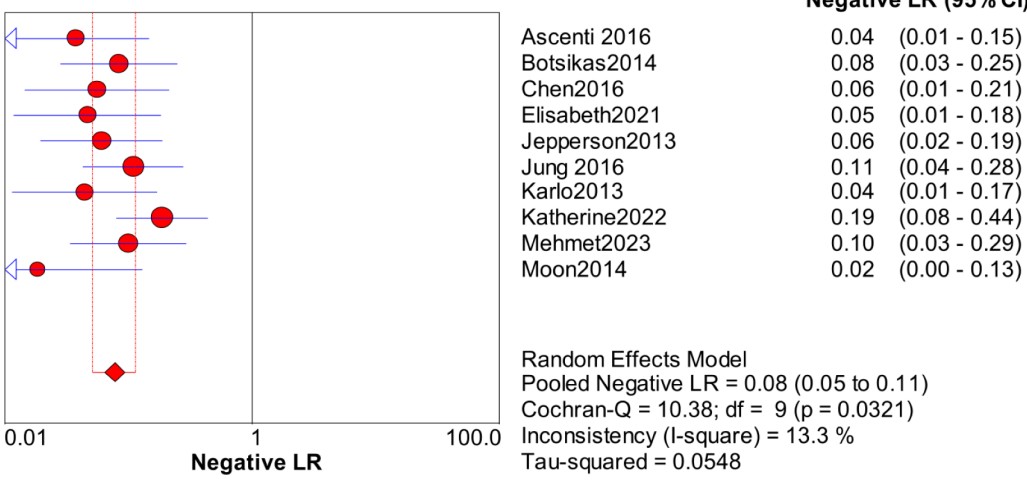

**Figure 5** Forest map of the NLR of dual energy CT in the detection of urinary calculi.

using the different sound and image characteristics of the different stone components to identify the composition of the stones before treatment. Because the CT values of different substances differ under different X-ray energy levels, the CT values of iodine agents vary significantly between the 80-kV and 140-kV images (*Tatsugami et al., 2022*). After data conversion, the iodized substances are removed from the enhanced scanning images, thus achieving material separation. A comparative investigation of DECT and CNCT in terms of image quality, tissue, organ structure, and CT values, amongst others, by *Sajja et al. (2020)* indicated that DECT could replace the diagnostic role of CNCT. In the CTU excretion phase images, urinary calculi are often indistinguishable from iodized contrast agents. Dual-energy CT can provide information about the size and location of urinary calculi by removing iodized substances from the CT images (*Schyns et al., 2017*). Thus, using a

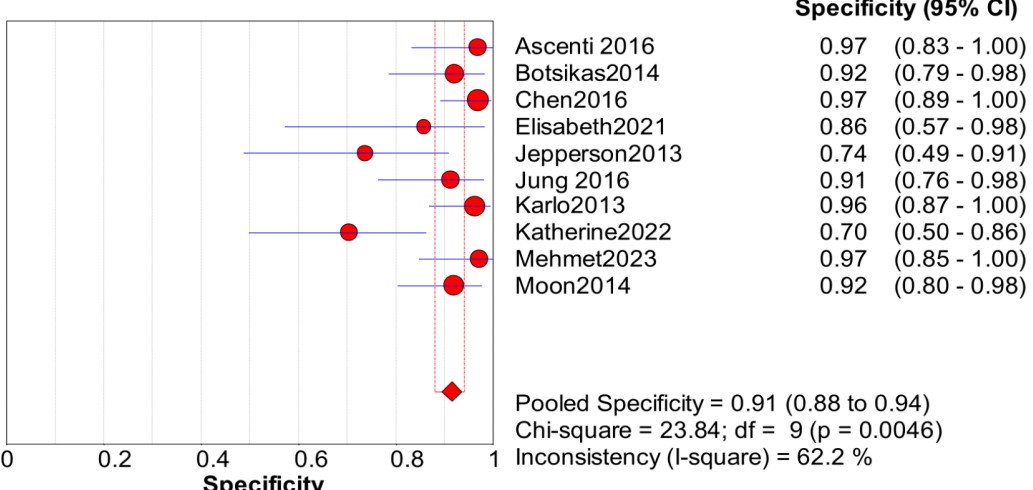

**Figure 6** Forest map of the specificity of dual energy CT in the detection of urinary calculi.

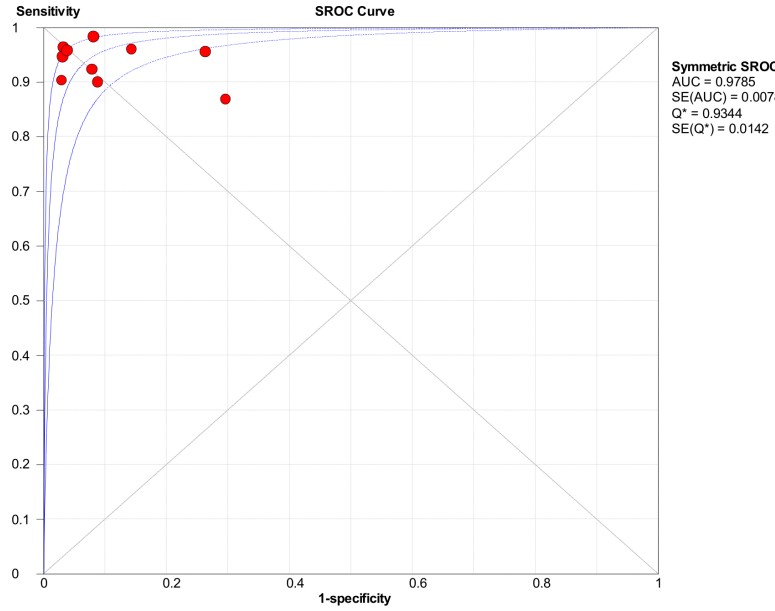

**Figure 7** ROC curve of dual energy VNCT detection in urinary calculi.

one-stop dual-energy CT examination, the information of both plain scan and CTU images can be obtained at the same time (*Fernández-Pérez et al., 2022*).

The image noise of dual-energy CT is higher than that of CNCT, and DECT requires more time and expertise to process image data and generate images (*Sodickson et al., 2021*). One of the factors limiting the broad application of DECT in CTU is the incomplete subtraction of the high concentration of iodized contrast medium excreted in the urine; the ''marginal artifact'' produced by this high concentration of iodized contrast agent can conceal the

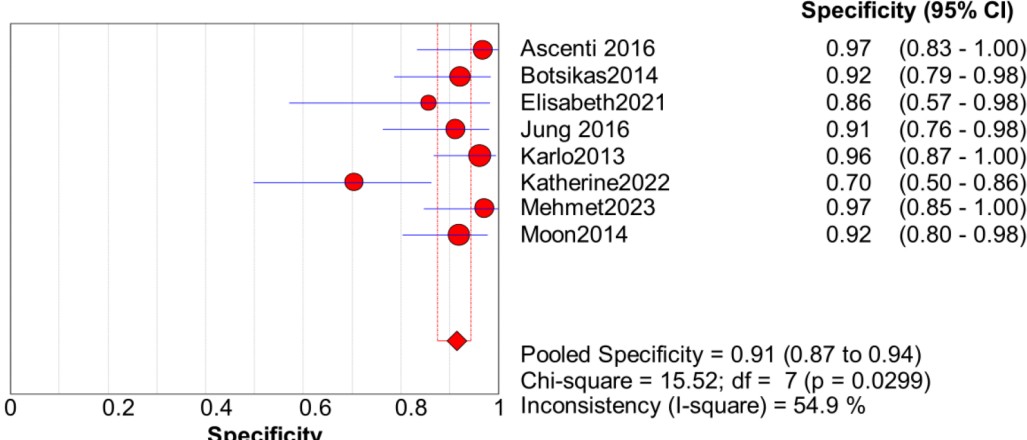

**Figure 8** Forest map of the specificity of dual energy CT in the detection of urinary calculi <3 mm.

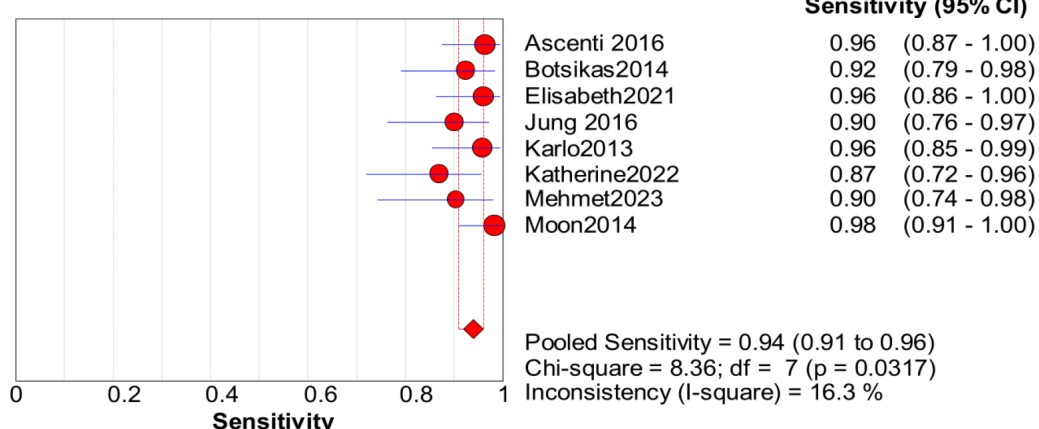

**Figure 9** Forest map of the sensitivity of dual energy CT in the detection of urinary calculi <3 mm.

**Table 2 Subgroup analysis.**

|  | Grouping | Sensitivity | Specificity | Meta regression | |
|---|---|---|---|---|---|
|  |  |  |  | P value | RDOR value |
| CT equipment | Siemens | 0.84 (0.81~0.87) | 0.97 (0.96~0.98) | 0.418 | 7.42 |
|  | GE | 0.90 (0.77~0.97) | 0.89 (0.56~1.00) |  |  |
| Tube voltage | 140 kV and 80 kV | 0.85 (0.81~0.88) | 0.97 (0.96~0.98) | 0.470 | 0.33 |
|  | 140 kV and 100 kV | 0.62 (0.51~0.72) | 0.98 (0.97~0.99) |  |  |
| Article quality | >3 points | 0.66 (0.58~0.74) | 0.96 (0.94~0.98) | 0.177 | 0.13 |
|  | <3 points | 0.93 (0.90~0.95) | 0.98 (0.96 ~0.98) |  |  |

**Notes.**
RDOR, relative diagnostic ratio.
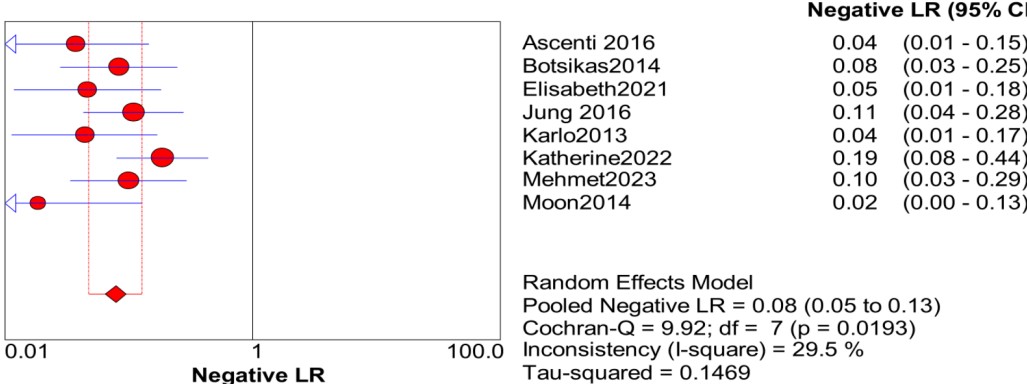

**Figure 10  Forest map of the NLR of dual energy CT in the detection of urinary calculi <3 mm.**

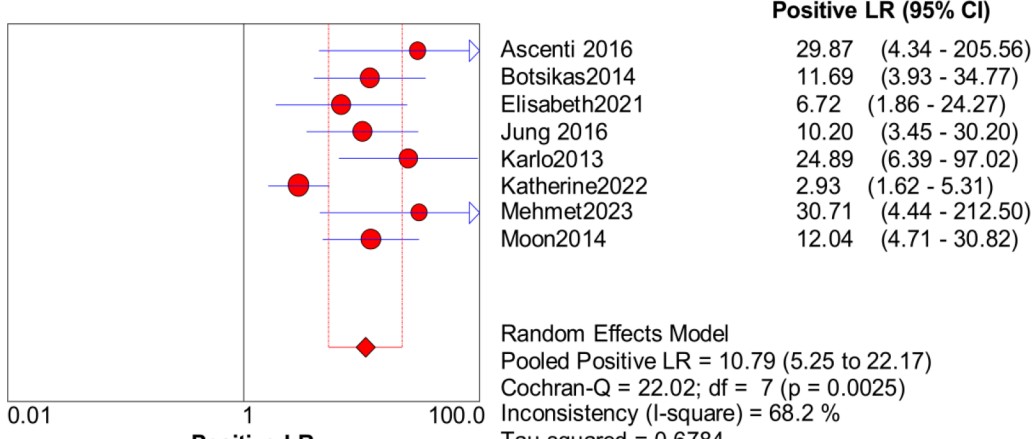

**Figure 11  Forest map of the sensitivity of PLR in the detection of urinary calculi <3 mm.**

stone, resulting in a false-negative result (*Xu et al., 2021*). In addition, the focus without deiodination may be mistaken for the stone, resulting in a false positive (*McCollough et al., 2015*). A total of 10 articles were included in this meta-analysis. The results of the meta-analysis showed that dual-energy CT had good sensitivity and specificity for the detection of urinary calculi, especially for urinary calculi <3 mm in diameter. In addition, subgroup and sensitivity analyses were performed to explore the source of heterogeneity. Firstly, we extracted the data from three subgroups, and no source of heterogeneity was observed in subgroup analysis. Moreover, the sensitivity analysis showed that the study by *Botsikas et al. (2014)* significantly impacted the heterogeneity. The reason for this may be related to the preliminary design of the study (the delay period occurred at 10 min after 100 mL administration, and the lack of observed small stones may have been due to attenuation of contrast medium in the urinary system). Furthermore, the number of articles included in the meta-analysis was relatively small, and there were differences in the

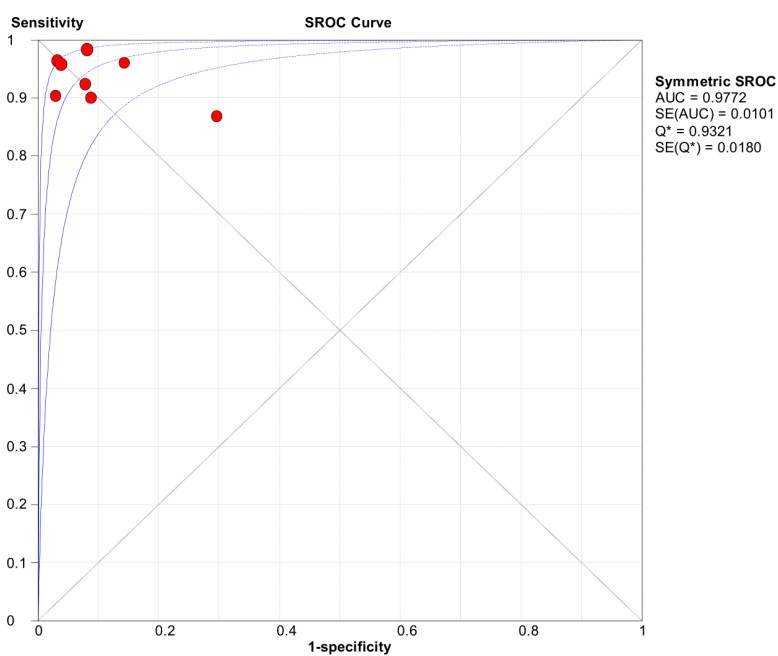

**Figure 12 ROC curve of dual energy VNCT detection in urinary calculi <3 mm.**

sample size and quality of the different articles, thus requiring additional confirmation by well-designed and high-quality studies in the future.

In the articles included in this study, the DECT phase was the delayed phase. There was no unified standard for studying the DECT phase, and the phases used by different researchers also differ. Thus, the detection sensitivity of urinary calculi with lengths <three mm was high (*Spek et al., 2016*). A previous study found that most ureteral stones with lengths of less than three mm can be excreted by themselves, although even if the length of the stone is less than three mm, it can cause pain and microscopic hematuria (*Singh et al., 2020*).

This meta-analysis has several strengths as well as some limitations. While the document retrieval was broad, it was limited to studies reported in English. There was slight heterogeneity between the included studies, which might have affected the results. The samples and diagnostic tests involved in the meta-analysis were relatively limited, so further multicenter studies with large samples are needed to clarify these factors.

## CONCLUSIONS

In brief, the meta-analysis showed that dual-energy CT had high diagnostic sensitivity and specificity in the diagnosis of urinary calculi. It could thus replace conventional CT in CTU examinations and reduce the radiation dose, suggesting its potential in clinical applications. After post-processing, dual-energy CT can produce enhanced scan images and iodine maps, which not only allow the detection of stone and their compositions but also provides more clinical information.

### Funding
The authors received no funding for this work.

### Competing Interests
The authors declare there are no competing interests.

### Author Contributions
- Peipei Feng conceived and designed the experiments, analyzed the data, prepared figures and/or tables, authored or reviewed drafts of the article, and approved the final draft.
- Guochao Li conceived and designed the experiments, performed the experiments, prepared figures and/or tables, and approved the final draft.
- Peng Liang performed the experiments, analyzed the data, authored or reviewed drafts of the article, and approved the final draft.

### Human Ethics
The following information was supplied relating to ethical approvals (*i.e.*, approving body and any reference numbers):

The study was approved by the Institutional Review Board and Research Ethics Committee of the Yantaishan Hospital.

### Data Availability
The raw data is available in the Supplemental File.

### Supplemental Information
Supplemental information for this article can be found online at http://dx.doi.org/10.7717/peerj.16076#supplemental-information.

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
