# Peer review of "The value of dual-energy computed tomography (DECT) in the diagnosis of urinary calculi: a systematic review and meta-analysis of retrospective studies"

_PeerJ, doi:10.7717/peerj.16076_

## Round 0.1 · original submission · Minor Revisions

Please carefully read the comments from all reviewers and provide your point-to-point responses to address them.

Reviewer 1 ·

Basic reporting

The manuscript is interesting and well-written. The authors fully introduced the background of the article and provided a detailed explanation of the conclusion of the paper

Experimental design

The manuscript is interesting and well-written. The value of dual-energy computed tomography (DECT) in the diagnosis of urinary calculi were investigated. The results are solid and indicate that patients in each subgroup should receive distinct personalized treatment. The manuscript can be considered for publication after addressing the following questions:
1. Abbreviations might be appeared in abstract. In abstract, full name is permanently a necessity.
2. The value “the value of DECT in the diagnosis of urinary calculi” in abstract conclusion differs from results: WHY?
3. The authors of this study referred to 10 articles for meta-analysis. However, most of the articles are from Asian countries, and authors are advised to add articles from other countries.
4. There is an omission in this work is ignoring conclusion’s part.
5. It is suggested to explore the source of heterogeneity with the results of subgroup analysis as supplement.
6. It will be better to show kappa for the selection and data extraction. Please show the data of kappa of agreement during the systematic searches.
7. Line 52, enhanced scanning of dual-energy CT in the diagnosis of urinary calculi could reduce the scanning times and reduces the effective radiation dose by 20% to 50%. Is there any innovation in this article compared to the previous article?
8. The data shown here are rather underwhelming, to be honest, and it is not clear on which basis studies/reports were included and why others (a lot of studies, in fact) have been excluded. It is described in materials & methods, how studies were included/excluded, and it is difficult for a reviewer to appreciate this without performing a lengthy literature study by himself.

Validity of the findings

.

Reviewer 2 ·

Basic reporting

(1) Please note that the reviewers have deemed the quality of language in this manuscript unsuitable for publication. Please send the manuscript to a language editing company to improve the article for language and style. Please provide the certificate confirming that language editing has been performed at the same time as the response to the peer review comments.
(2) I strongly recommend that the authors add strength, limitation, and conclusion at the end of this manuscript.
(3) In short, I suggest including studies from all over the world, not just Asia, provided these are available. If not, the authors must very clearly state why there is a location bias or population bias, and which consequences this may have for the scientific relevance of the manuscript.
(4) The method of this paper is not innovative enough. Authors need to highlight their innovative contributions. In general, there is some lack of explanation of methods used in the study.

Experimental design

.

Validity of the findings

.

Reviewer 3 ·

Basic reporting

Good

Experimental design

Good

Validity of the findings

Good

Additional comments

I recommend minor revision for the manuscript according to the following issues.
1. In the methods part, please describe the participants, intervention (exposure), comparison, outcome, and study design according to the principle of PICO(S).
2. The resolution of the figures should be improved. Besides, Figure 1 should be prepared according to the standard PRISMA guidance.
3. How about the results of publication biases detection?
4. In the abstract, the 95% CI of each pooled indicator should be provided.

---

## Round 0.2 · Minor Revisions

The English language quality is poor and the manuscript needs to be edited. You may (i) have a colleague who is proficient in English and familiar with the subject matter review the manuscript, or (ii) contact a professional editing service to review the manuscript. PeerJ can provide language editing services - you can contact PeerJ at copyediting@peerj.com for pricing (be sure to provide your manuscript number and title).

**Language Note:** The Academic Editor has identified that the English language must be improved. PeerJ can provide language editing services - please contact us at copyediting@peerj.com for pricing (be sure to provide your manuscript number and title). Alternatively, you should make your own arrangements to improve the language quality and provide details in your response letter. – PeerJ Staff

Reviewer 3 ·

Basic reporting

No any more comments.

Experimental design

No any more comments.

Validity of the findings

No any more comments.

Additional comments

The quality of the manuscript is improved after revision. It could be accepted for publication.

---

## Round 0.3 · accepted · Accept

The language has been improved in the revised paper.